# Antimicrobial prescription practices for outpatients with acute respiratory tract infections: A retrospective, multicenter, medical record-based study

**Tomoharu Ishida, Hideharu Hagiya**◉*, **Hiroyuki Honda, Yasuhiro Nakano, Hiroko Ogawa, Mikako Obika, Keigo Ueda, Hitomi Kataoka, Yoshihisa Hanayama, Fumio Otsuka**◉

Department of General Medicine, Okayama University Graduate School of Medicine, Dentistry and Pharmaceutical Sciences, Kitaku, Okayama, Japan

* hagiya@okayama-u.ac.jp

**Data Availability Statement:** Data for this study are within the Supporting Information files.

## Abstract

Antimicrobial stewardship for outpatients with acute respiratory tract infections (ARTIs) should be urgently promoted in this era of antimicrobial resistance. Previous large-sample studies were based on administrative data and had limited reliability. We aimed to identify current antimicrobial prescription practices for ARTIs by directly basing on medical records. This multicenter retrospective study was performed from January to December in 2018, at five medical institutes in Japan. We targeted outpatients aged ≥18 years whose medical records revealed International Classification of Diseases (ICD-10) codes suggesting ARTIs. We divided the eligible cases into three age groups (18–64 years, 65–74 years, and ≥75 years). We defined broad-spectrum antimicrobials as third-generation cephalosporins, macrolides, fluoroquinolones, and faropenem. Primary and secondary outcomes were defined as the proportion of antimicrobial prescriptions for the common cold and other respiratory tract infections, respectively. Totally, data of 3,940 patients were collected. Of 2,914 patients with the common cold, 369 (12.7%) were prescribed antimicrobials. Overall, compared to patients aged ≥75 years (8.5%), those aged 18–64 years (16.6%) and those aged 65–74 years (12.1%) were frequently prescribed antimicrobials for the common cold (odds ratio [95% confidential interval]; 2.15 [1.64–2.82] and 1.49 [1.06–2.09], respectively). However, when limited to cases with a valid diagnosis of the common cold by incorporating clinical data, no statistical difference was observed among the age groups. Broad-spectrum antimicrobials accounted for 90.2% of the antimicrobials used for the common cold. Of 1,026 patients with other respiratory infections, 1,018 (99.2%) were bronchitis, of which antimicrobials were prescribed in 49.9% of the cases. Broad-spectrum antimicrobials were the main agents prescribed, accounting for nearly 90% of prescriptions in all age groups. Our data suggested a favorable practice of antimicrobial prescription for outpatients with ARTIs in terms of prescribing proportions, or quantitative aspect. However, the prescriptions were biased towards broad-spectrum antimicrobials, highlighting the need for further antimicrobial stewardship in the outpatient setting from a qualitative perspective.

**Funding:** This research did not receive any specific grant from funding agencies in the public, commercial, or not-for-profit sectors.

**Competing interests:** The authors have declared that no competing interests exist.

## Introduction

The emergence of antimicrobial resistance (AMR) is a global threat to public health, suggesting an urgent need for antimicrobial stewardship (AMS) [1]. To fight against this critical situation worldwide, the World Health Organization issued a statement in 2015 to initiate strategies to reduce the risk of this transmission hazard [2]. Thereafter, the Group of Seven (G7) countries formulated 10-year action plans [3], in which the promotion of AMS was intensively highlighted. In Japan, the National Action Plan on Antimicrobial Resistance was launched in 2016, which proved to be a cornerstone for the optimization of antimicrobial prescription practices [4].

Reduction of inappropriate and unnecessary antimicrobial prescriptions for acute respiratory tract infections (ARTIs) is the most important part of AMS. ARTIs are one of the most common outpatient infections, most (>90%) of which involves viral etiology [5,6] and thus do not require antimicrobial treatment. According to the guidelines, prescription of antimicrobials is often unnecessary for ARTIs treatment, although there are actually cases that require a delayed prescription [7,8]. However, in reality, we, as Japanese clinicians, frequently witness antimicrobials being prescribed in outpatient settings, partially because of requests from patients or due to physicians' anxiety. According to a study conducted in 2009 using health insurance claims data submitted to an employer-sponsored plan, antimicrobials were prescribed for approximately 60% of non-bacterial ARTIs in Japan [9]. A Japanese nationwide population-based study, which also used health insurance claims data, reported that more than 94% of the antimicrobials administered during 2011–2013 were in the oral form, mostly for the outpatients [10]. Another study, which was based on health insurance claims data of 8.65 million visits, revealed that the physician visit rate for patients with ARTIs was 990.6 (99% confidence interval [CI], 989.4–991.7) per 1000 person-years, equivalent to one visit per year for each individual in Japan, and antimicrobials were prescribed in approximately half of these visits (532.4 per 1000 person-years; 99% CI, 531.6–533.3) [11]. Most of the antimicrobials prescribed for ARTIs were broad-spectrum oral formulations including cephalosporins (41.9%, of which third-generation cephalosporins accounted for 97.3%), macrolides (32.8%), and fluoroquinolones (14.7%) [12]. A similar study using a health insurance claims database was conducted among the pediatric population in Japan, revealing that broad-spectrum cephalosporins (38.3%) and macrolides (25.8%) were frequently administered to preschool children with ARTIs [13].

Despite the higher antimicrobial prescription rates in Japan, downward trends have been reported. A retrospective, observational study using longitudinal, administrative claims data revealed that a mean monthly antimicrobial prescription rate for nonbacterial-ARTIs was 31.65 per 100 visits between April 2012 and June 2017 [14]. The antimicrobial prescription rate decreased by 19.2% during the study period; however, there was no remarkable trend change compared to other countries. For instance, previous national data in the United States (1995 to 2006) suggested that ARTIs-associated antimicrobial prescriptions decreased by 36% among children younger than 5 years and by 18% among persons aged 5 years or older [15]. According to another analysis of nationally representative data in the United States (2000 to 2010), antimicrobial prescription for ARTIs decreased by 57% among children and adolescents (<18 years) and 38% among adults (18 to 64 years), although there was no certain trend among those aged ≥65 years [16]. Thus, it is possible that the decrease in the proportion of antimicrobial prescriptions for ARTIs in Japan may be further accelerated.

To date, the administrative data have revealed over-prescriptions of antimicrobials for ARTIs in Japan. However, the health insurance claims data inevitably lack credibility because they are accumulated without clinical records. Therefore, it is unclear whether these results

reflect the actual status of antimicrobial prescription practices. Currently, there is a need for medical record-based data analysis to improve the reliability of the data. The present study aimed to determine the proportion of antimicrobial prescriptions for ARTIs, especially for the common cold, by directly examining the medical records.

## Materials and methods

### Study population, period, and subjects

This was a multicenter retrospective study of patients who visited the outpatient clinics of five medical institutions in Okayama and Kagawa prefectures in Japan (Marugame Medical Center, Kasaoka City Hospital, Tamano City Hospital, Kaneda Hospital, and Niimi National Health Insurance Clinics [Kojiro Clinic, Niizato Clinic, and Yukawa Clinic]). All these institutes are located in the rural areas, and the patient population is almost identical. The first four institutes (Marugame, Kasaoka, Tamano, and Kaneda) are regional general hospitals with inpatient beds, while the Niimi National Health Insurance Clinics are no-bedded outpatient clinics. We included outpatients aged ≥18 years whose medical records included International Classification of Diseases (ICD-10) codes suggesting ARTIs between January 1, 2018, and December 31, 2018.

### Definition of ARTIs

We used the ICD-10 codes to define ARTIs as follows: acute nasopharyngitis (J00), acute sinusitis (J01), acute pharyngitis (J02), acute tonsillitis (J03), acute laryngitis and tracheitis (J04), acute obstructive laryngitis and epiglottitis (J05), ARTIs at multiple and unspecified sites (J06), acute bronchitis (J20–22), bronchitis specified as neither acute nor chronic (J40), acute upper respiratory tract infection (J069), and acute bronchitis without details (J209), by referring to previous literature [6,11,17]. Of these codes, we classified J01 as nasal; J02, J03, J04, and J05 as pharyngeal; and J20, J21, J22, J40, and J209 as lower respiratory codes. Patients with common cold were defined as those with the codes J00, J06, and J069 and those with codes for infection in two or more regions of the respiratory tract.

### Data collection

We collected data on age, sex, presence of upper or lower respiratory symptoms (nasal [nasal discharge, nasal obstruction], pharyngo-laryngeal [sore throat, hoarseness of voice], and bronchial [cough, sputum expectoration] regions), clinical diagnosis defined by the ICD-10 codes, and antimicrobial prescriptions from medical records. In Japan, only medical doctors are authorized to prescribe antimicrobials, and not nurse practitioners or other healthcare professionals. The antimicrobial prescriptions included in this study were not limited to either general practitioners or organ specialists. Patients were categorized into three age groups for the analysis: 18–64 years, 65–74 years, and ≥75 years. We excluded patients who revisited the outpatient department within 30 days from the first visit and those who received intravenous antimicrobial therapy; *i.e.*, our study included patients prescribed oral antimicrobials alone. As reported in previous studies [15,18], we considered third-generation cephalosporins, macrolides, fluoroquinolones, and faropenem as broad-spectrum antimicrobials, while we considered penicillins as narrow-spectrum antimicrobials.

### Outcomes and statistical analysis

We stratified the eligible cases by the presence of clinical manifestations associated with ARTIs. Cases were grouped by the involvement of respiratory tract regions (nasal, pharyngo-

laryngeal, and bronchial), with descriptions of three, two, and two or more regions involved. Categorical variables were shown in the numbers, percentages, and odds ratios (ORs) with their 95% confidence intervals (CIs), which were assessed with the chi-squared test or Fisher's exact test, as appropriate. Continuous variables were summarized with median and interquartile range (IQR). The primary outcome was the proportion of antimicrobial prescriptions and the drugs prescribed for patients diagnosed with the common cold. The secondary outcomes were defined as the proportion of antimicrobial prescriptions and the drugs prescribed for patients diagnosed with respiratory tract infections other than the common cold. The data were analyzed using EZR software, a graphic user interface for the R 3.5.2 software (The R Foundation for Statistical Computing, Vienna, Austria). All reported *p* values less than 0.05 were considered statistically significant.

### Ethics approval, funding and conflict of interests

Informed consent was not necessary because the data were anonymized. The study was approved by the Okayama University's Graduate School of Medicine, Dentistry and Pharmaceutical Sciences and Okayama University Hospital's Ethics Committee (No. 1907–036).

### Results

We collected data of 3,955 patients from the five institutions. Fifteen patients received intravenous antimicrobial therapy and were thus excluded. Finally, the data of 3,940 patients were analyzed: Marugame Medical Center (n = 996), Kasaoka City Hospital (n = 1433), Tamano City Hospital (n = 818), Kaneda Hospital (n = 642), and Niimi National Health Insurance Clinics (n = 51) (Fig 1). The median patient age [IQR] was 68 [47, 79] years. The numbers and percentages of patients in each age group was as follows: 1,684 (42.7%) aged 18–64 years, 818 (20.8%) aged 65–74 years, and 1,438 (36.5%) aged ≥75 years. The background data as well as the number of patients for each ICD-10 code of the included cases in each medical institute was given in Table 1.

The number and proportions (*i.e.*, quantity) of antimicrobial prescriptions for the common cold are summarized in Table 2. The total number of patients diagnosed with the common cold was 2,914, of which antimicrobials were prescribed in 369 cases (12.7%). By age, 16.6% of patients aged 18–64 years were prescribed antimicrobials, which was statistically higher than

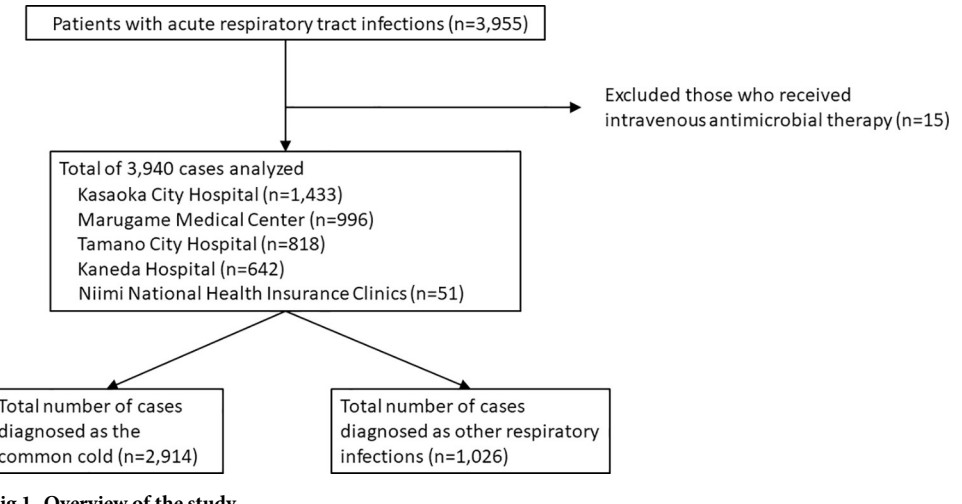

**Fig 1. Overview of the study.**

**Table 1. Numbers and percentages of background data and diagnosis of the cases in each medical institute by age groups.**

| | Overall | Kasaoka | Marugame | Tamano | Kaneda | Niimi |
|---|---|---|---|---|---|---|
| | | | | Medical institutes | | |
| **The number of cases (%)** | | | | | | |
| Overall | 3,940 | 1,433 (36.4) | 996 (25.3) | 818 (20.8) | 642 (16.3) | 51 (1.3) |
| 18–64 years | 1,684 (42.7) | 456 (27.1) | 594 (35.3) | 313 (18.6) | 310 (18.4) | 11 (0.7) |
| 65–74 years | 818 (20.8) | 297 (36.3) | 179 (21.9) | 184 (22.5) | 148 (18.1) | 10 (1.2) |
| ≥75 years | 1,438 (36.5) | 680 (47.3) | 223 (15.5) | 321 (22.3) | 184 (12.8) | 30 (2.1) |
| Median age [IQR], years | 68 [47, 79] | 74 [60, 81] | 57 [38, 73] | 70 [49, 79] | 65 [44.5, 76] | 75.5 [67.5, 83] |
| Sex (F/M) (%) | 2,257/1,683 | 888/545 | 536/460 | 487/331 | 316/326 | 30/21 |
| | (57.3/42.7) | (62.0/38.0) | (53.8/46.2) | (59.5/40.5) | (49.2/50.8) | (58.8/41.2) |
| **ICD-10 codes (%)** | | | | | | |
| [J00] | 576 | 61 (10.6) | 52 (9.0) | 253 (43.9) | 157 (27.3) | 51 (8.9) |
| [J01] | 6 | 2 (33.3) | 1 (16.7) | 3 (50.0) | 0 | 0 |
| [J02] | 306 | 98 (32.0) | 84 (27.5) | 81 (26.5) | 43 (14.1) | 0 |
| [J03] | 38 | 0 | 19 (50.0) | 18 (47.4) | 1 (2.6) | 0 |
| [J04] | 4 | 0 | 1 (25.0) | 1 (25.0) | 1 (25.0) | 0 |
| [J05] | 0 | 0 | 0 | 0 | 0 | 0 |
| [J06] | 9 | 1 (11.1) | 3 (33.3) | 4 (44.4) | 1 (11.1) | 0 |
| [J20] | 0 | 0 | 0 | 0 | 0 | 0 |
| [J21] | 0 | 0 | 0 | 0 | 0 | 0 |
| [J22] | 0 | 0 | 0 | 0 | 0 | 0 |
| [J40] | 160 | 23 (14.4) | 13 (8.1) | 69 (43.1) | 56 (35.0) | 0 |
| [J069] | 2,244 | 1,095 (48.8) | 630 (28.1) | 187 (8.3) | 332 (14.8) | 0 |
| [J209] | 1,107 | 277 (25.0) | 351 (31.7) | 368 (33.2) | 111 (10.0) | 0 |

The International Classification of Diseases (ICD-10) codes were endorsed in May 1990 by the Forty-third World Health Assembly to develop the diagnostic classification standard for all clinical and research purposes. IQR, interquartile range. Definitions of each disease to ICD-10 code were as follows: Acute nasopharyngitis (J00), acute sinusitis (J01), acute pharyngitis (J02), acute tonsillitis (J03), acute laryngitis and tracheitis (J04), acute obstructive laryngitis and epiglottitis (J05), acute respiratory tract infections s at multiple and unspecified sites (J06), acute bronchitis (J20–22), bronchitis specified as neither acute nor chronic (J40), acute upper respiratory tract infection (J069), and acute bronchitis without details (J209).

that seen in patients aged ≥75 years (8.5%; OR [95% CI], 2.15 [1.64–2.82]; *p* value <0.001). Similarly, 12.1% of patients aged 65–74 years were prescribed antimicrobials, which was significantly higher than that of patients aged ≥75 years (OR [95% CI], 1.49 [1.06–2.09]; *p* value = 0.02). Comparing the antimicrobial prescription between those aged 18–64 years and those aged 65–74 years, the prescription proportion was significantly higher in the younger group (OR [95% CI], 1.45 [1.08–1.96]; *p* value = 0.012). These figures included all the collected cases from the record review, without incorporating data on the presence or absence of descriptions relating to ARTIs.

We then stratified the cases according to the number of clinical manifestations suggestive of ARTIs. The proportion of antimicrobial prescriptions in "three respiratory regions", "two respiratory regions", and "two or more respiratory regions" were 16.7% (43/258), 18.2% (117/644), and 17.7% (160/902), respectively. In the "three respiratory regions" group, the antimicrobial prescriptions in the 18–64 years age group (18.5%) and 65–74 years age group (17.1%) were higher than that of ≥75 years age group (8.9%). However, the statistical analysis did not reveal any significant difference among the groups. Similarly, in the "two respiratory regions" and "two or more respiratory regions" cohorts, there was no significant difference observed in the antimicrobial prescriptions among the age groups (Table 2).

**Table 2. Numbers and proportions of antimicrobial prescriptions for common cold according to clinical manifestations confirmed in the medical records and age groups.**

| | Visits | Antimicrobial prescription | | vs ≥75 years | | vs 65–74 years | |
| --- | --- | --- | --- | --- | --- | --- | --- |
| | Numbers | Numbers | % (95% CI) | Odds ratio (95% CI) | p values | Odds ratio (95% CI) | p values |
| **Total (clinical data not incorporated)** | | | | | | | |
| overall | 2914 | 369 | 12.7% (11.5–13.9) | - | - | - | - |
| 18–64 years | 1243 | 206 | 16.6% (14.5–18.8) | 2.15 (1.64–2.82) | <0.001 | 1.45 (1.08–1.96) | 0.012 |
| 65–74 years | 596 | 72 | 12.1% (9.6–15.0) | 1.49 (1.06–2.09) | 0.02 | Reference | |
| ≥75 years | 1075 | 91 | 8.5% (6.9–10.3) | Reference | | ND | |
| **Three respiratory regions*** | | | | | | | |
| overall | 258 | 43 | 16.7% (12.3–21.8) | - | - | - | - |
| 18–64 years | 178 | 33 | 18.5% (13.1–25.0) | 2.33 (0.76–9.56) | 0.18 | 1.10 (0.40–3.50) | 1 |
| 65–74 years | 35 | 6 | 17.1% (6.6–33.6) | 2.10 (0.45–11.07) | 0.32 | Reference | |
| ≥75 years | 45 | 4 | 8.9% (2.5–21.2) | Reference | | ND | |
| **Two respiratory regions**** | | | | | | | |
| overall | 644 | 117 | 18.2% (15.3–21.4) | - | - | - | - |
| 18–64 years | 358 | 67 | 18.7% (14.8–23.1) | 1.09 (0.66–1.85) | 0.81 | 1.08 (0.62–1.93) | 0.89 |
| 65–74 years | 125 | 22 | 17.6% (11.4–25.4) | 1.01 (0.52–1.96) | 1 | Reference | |
| ≥75 years | 161 | 28 | 17.4% (11.9–24.1) | Reference | | ND | |
| **Two or more respiratory regions***** | | | | | | | |
| overall | 902 | 160 | 17.7% (15.3–20.4) | - | - | - | - |
| 18–64 years | 536 | 100 | 18.7% (15.4–22.2) | 1.25 (0.80–1.99) | 0.34 | 1.08 (0.67–1.79) | 0.82 |
| 65–74 years | 160 | 28 | 17.5% (12.0–24.3) | 1.15 (0.63–2.09) | 0.67 | Reference | |
| ≥75 years | 206 | 32 | 15.5% (10.9–21.2) | Reference | | ND | |

CI, confidence interval; ND, no data.

*"Three respiratory regions" denotes that clinical manifestations of all three distinct respiratory tract regions (nasal, pharyngo-laryngeal, and bronchial) were described in the medical records.

**"Two respiratory regions" and

***"Two or more respiratory regions" denote that as it appears, clinical manifestations of each of the two and two or more of the respiratory tract regions were described.

Quality of antimicrobial prescription for the common cold, which was evaluated by the prescription proportions of broad-spectrum antimicrobials are provided in Table 3. Overall, broad-spectrum agents accounted for 90.2% of antimicrobials selected for the common cold. High prescription rates were also observed even when incorporating the clinical data that validates the diagnosis of the common cold; "three respiratory regions" (97.7%), "two respiratory regions" (84.6%), and "two or more respiratory regions" (88.1%). Oral forms of penicillins were selected in less than 10% of the common cold cases in any conditions. This prescribing trend of being extremely biased toward broad-spectrum antimicrobials was also observed when stratifying the cases into each age group (Table 2).

The proportions of antimicrobial prescriptions and prescribed drugs for other respiratory infections, such as sinusitis, pharyngitis, tonsillitis, laryngitis, and bronchitis, were summarized in Table 4. The total number of visits was 1,026, and the overall proportion of visits resulting in antimicrobial prescriptions was 50.1%. Bronchitis accounted for most of the cases (99.2%), and nearly half of them received antimicrobial treatment: 55.0% in 18–64 years, 44.1% in 65–74 years, and 47.1% in ≥75 years. Broad-spectrum antimicrobials were the main agents prescribed, accounting for nearly 90% of prescriptions for all age categories.

**Table 3. Proportions of antimicrobials prescribed for common cold according to age groups and clinical manifestations.**

| | Broad-spectrum antimicrobials[††] | | | | Other antimicrobials | |
|---|---|---|---|---|---|---|
| | 3rd-cephem[†] | Macrolides | Fluoroquinolones | Faropenem | Penicillins | Miscellaneous |
| **Overall** | | | | | | |
| Total[§] | 90.2 (86.7–93.1) | | | | 9.8 (6.9–13.3) | |
| | 23.6 (19.3–28.2) | 41.2 (36.1–46.4) | 24.1 (19.8–28.8) | 1.4 (0.4–3.1) | 6.0 (3.8–8.9) | - |
| Three respiratory regions* | 97.7 (87.7–99.9) | | | | 2.3 (0.1–12.3) | |
| | 32.6 (19.1–48.5) | 46.5 (31.2–62.3) | 14.0 (5.3–27.9) | 4.7 (0.6–15.8) | 0 | - |
| Two respiratory regions** | 84.6 (76.8–90.6) | | | | 15.4 (9.4–23.2) | |
| | 17.1 (10.8–25.2) | 43.6 (34.4–53.1) | 22.2 (15.1–30.8) | 1.7 (0.2–6.0) | 9.4 (4.8–16.2) | - |
| Two or more respiratory regions*** | 88.1 (82.1–92.7) | | | | 11.9 (7.3–17.9) | |
| | 21.3 (15.2–28.4) | 44.4 (36.5–52.4) | 20 (14.1–27) | 2.5 (0.7–6.3) | 6.9 (3.5–12) | - |
| **18–64 years** | | | | | | |
| Total[§] | 89.8 (84.8–93.6) | | | | 10.2 (6.4–15.2) | |
| | 27.7 (21.7–34.3) | 34 (27.5–40.9) | 27.2 (21.2–33.8) | 1.0 (0.1–3.5) | 5.3 (2.7–9.4) | - |
| Three respiratory regions* | 97.0 (84.2–99.9) | | | | 0 | |
| | 27.3 (13.3–45.5) | 48.5 (30.8–66.5) | 18. 2 (7–35.5) | 3.0 (0.1–15.8) | 0 | - |
| Two respiratory regions** | 83.6 (72.5–91.5) | | | | 16.4 (8.5–27.5) | |
| | 19.4 (10.8–30.9) | 44.8 (32.6–57.4) | 17.9 (9.6–29.2) | 1.5 (0–8.0) | 7.5 (2.5–16.6) | - |
| Two or more respiratory regions*** | 88.0 (80–93.6) | | | | 12 (6.4–20) | |
| | 22 (14.3–31.4) | 46 (36–56.3) | 18 (11–26.9) | 2.0 (0.2–7.0) | 5 (1.6–11.3) | - |
| **65–74 years** | | | | | | |
| Total[§] | 91.7 (82.7–96.9) | | | | 8.3 (3.1–17.3) | |
| | 22.2 (13.3–33.6) | 52.8 (40.7–64.7) | 15.3 (7.9–25.7) | 1.4 (0–7.5) | 6.9 (2.3–15.5) | - |
| Three respiratory regions* | 100 (54.1–100) | | | | 0 | |
| | 33. 3 (4.3–77.7) | 50 (11.8–88.2) | 0 | 16.7 (0.4–64.1) | 0 | - |
| Two respiratory regions** | 77.3 (54.6–92.2) | | | | 22.7 (7.8–45.4) | |
| | 18.2 (5.2–40.3) | 36.4 (17.2–59.3) | 22.7 (7.8–45.4) | 0 | 22.7 (7.8–45.4) | - |
| Two or more respiratory regions*** | 82.1 (63.1–93.9) | | | | 17.9 (6.1–36.9) | |
| | 21.4 (8.3–41) | 39.3 (21.5–59.4) | 17.9 (6.1–36.9) | 3.6 (0.1–18.3) | 17.9 (6.1–36.9) | - |
| **≥75 years** | | | | | | |
| Total[§] | 90.1 (82.1–95.4) | | | | 9.9 (4.6–17.9) | |
| | 15.4 (8.7–24.5) | 48.4 (37.7–59.1) | 24.2 (15.8–34.3) | 2.2 (0.3–7.7) | 6.6 (2.5–13.8) | - |
| Three respiratory regions* | 100 (39.8–100) | | | | 0 | |
| | 75 (19.4–99.4) | 25 (0.6–80.6) | 0 | 0 | 0 | - |
| Two respiratory regions** | 92.9 (76.5–99.1) | | | | 7.1 (0.9–2.4) | |
| | 10.7 (2.3–28.2) | 46.4 (27.5–66.1) | 32.1 (15.9–52.4) | 3.6 (0.1–18.3) | 3.6 (0.1–18.3) | - |
| Two or more respiratory regions*** | 93.8 (79.2–99.2) | | | | 6.2 (0.8–20.8) | |
| | 18.8 (7.2–36.2) | 43.8 (26.4–62.3) | 28.1 (13.7–46.7) | 3.1 (0.1–16.2) | 3.1 (0.1–16.2) | - |

Each proportion is given in percentages and 95% confidence intervals in total prescriptions.

[§]"Total" includes all the cases collected without incorporating data on the presence or absence of clinical manifestations.

*"Three respiratory regions" denotes that clinical manifestations of all three distinct respiratory tract regions (nasal, pharyngo-laryngeal, and bronchial) were described in the medical records.

**"Two respiratory regions" and

***"Two or more respiratory regions" denote that as it appears, clinical manifestations of each of the two and two or more of the respiratory tract regions were described.

[†] Third-generation cephalosporins.

[††]Total number of broad-spectrum antimicrobials and percentages of all prescribed antimicrobials.

**Table 4. Proportions of antimicrobial prescriptions and prescribed drugs for acute respiratory tract infections other than the common cold.**

| | Visits | No. and proportion of antimicrobial prescription | | % (95% CI) in total prescriptions | | | | | |
| --- | --- | --- | --- | --- | --- | --- | --- | --- | --- |
| | | | | Broad-spectrum antimicrobials[††] | | | | Other antimicrobials | |
| | No. | No. | % (95% CI) | 3rd-cephem[†] | Macrolides | Fluoroquinolones | Faropenem | Penicillins | Miscellaneous |
| **Overall** | 1026 | 514 | 50.1 (47.0–53.2) | 90.7 (87.8–93) | | | | 9.3 (7–12.2) | |
| | | | | 8.9 (6.6–11.8) | 43 (38.7–47.4) | 37.5 (33.3–41.9) | 1.2 (0.4–2.5) | 5.4 (3.6–7.8) | - |
| Sinusitis | 0 | - | - | - | | | | - | |
| | | | | - | - | - | - | - | - |
| Pharyngitis | 5 | 4 | 80 (28.4–99.5) | 75 (19.4–99.4) | | | | 25 (0.6–80.6) | |
| | | | | 50 (6.8–93.2) | 25 (0.6–80.6) | 0 | 0 | 25 (0.6–80.6) | - |
| Tonsillitis | 0 | - | - | - | | | | - | |
| | | | | - | - | - | - | - | - |
| Laryngitis | 3 | 2 | 66.7 (9.4–99.2) | 100 (15.8–100) | | | | - | |
| | | | | 50 (1.3–98.7) | 0 | 50(1.3–98.7) | 0 | - | - |
| Bronchitis | 1018 | 508 | 49.9 (46.8–53.0) | 90.7 (87.9–93.1) | | | | 9.3 (6.9–12.1) | |
| | | | | 8.5 (6.2–11.2) | 43.3 (38.9–47.7) | 37.8 (33.6–42.2) | 1.2 (0.4–2.6) | 5.3 (3.5–7.6) | - |
| **18–64 years** | 441 | 245 | 55.6 (50.8–60.3) | 91.0 (86.7–94.3) | | | | 9.0 (5.7–13.3) | |
| | | | | 8.6 (5.4–12.8) | 44.9 (38.6–51.4) | 36.3 (30.3–42.7) | 1.2 (0.3–3.5) | 4.5 (2.3–7.9) | - |
| Sinusitis | 0 | - | - | - | | | | - | |
| | | | | - | - | - | - | - | - |
| Pharyngitis | 4 | 4 | 100 (39.8–100) | 75 (19.4–99.4) | | | | 25 (0.6–80.6) | |
| | | | | 50 (6.8–93.2) | 25 (0.6–80.6) | 0 | 0 | 25 (0.6–80.6) | - |
| Tonsillitis | 0 | - | - | - | | | | - | |
| | | | | - | - | - | - | - | - |
| Laryngitis | 1 | 1 | 100 (2.5–100) | 100 (2.5–100) | | | | 0 | |
| | | | | 0 | 0 | 100 (2.5–100) | 0 | 0 | 0 |
| Bronchitis | 436 | 240 | 55.0 (50.2–59.8) | 91.2 (86.9–94.5) | | | | 8.8 (5.5–13.1) | |
| | | | | 7.9 (4.8–12.1) | 45.4 (39–51.9) | 36.7 (30.6–43.1) | 1.3 (0.3–3.6) | 4.2 (2–7.5) | - |
| **65–74 years** | 222 | 98 | 44.1 (37.5–50.9) | 88.8 (80.8–94.3) | | | | 11.2 (5.7–19.2) | |
| | | | | 8.2 (3.6–15.5) | 44.9 (34.8–55.3) | 34.7 (25.4–45) | 1.0 (0–5.6) | 7.1 (2.9–14.2) | - |
| Sinusitis | 0 | - | | - | | | | - | |
| | | | | - | - | - | - | - | - |
| Pharyngitis | 0 | - | | - | | | | - | |
| | | | | - | - | - | - | - | - |
| Tonsillitis | 0 | - | | - | | | | - | |
| | | | | - | - | - | - | - | - |
| Laryngitis | 0 | - | | - | | | | - | |
| | | | | - | - | - | - | - | - |
| Bronchitis | 222 | 98 | 44.1 (37.5–50.9) | 88.8 (80.8–94.3) | | | | 11.2 (5.7–19.2) | |
| | | | | 8.2 (3.6–15.5) | 44.9 (34.8–55.3) | 34.7 (25.4–45) | 1.2 (0.4–2.5) | 7.1 (2.9–14.2) | |
| **≥75 years** | 363 | 171 | 47.1 (41.9–52.4) | 91.2 (85.9–95) | | | | 8.8 (5–14.1) | |
| | | | | 9.9 (5.9–15.4) | 39.2 (31.8–46.9) | 40.9 (33.5–48.7) | 1.2 (0.1–4.2) | 5.8 (2.8–10.5) | - |
| Sinusitis | 0 | - | - | - | | | | - | |
| | | | | - | - | - | - | - | - |
| Pharyngitis | 1 | 0 | 0 | - | | | | - | |
| | | | | - | - | - | - | - | - |
| Tonsillitis | 0 | - | - | - | | | | - | |
| | | | | - | - | - | - | - | - |

(*Continued*)

**Table 4.** (Continued)

| | Visits | No. and proportion of antimicrobial prescription | % (95% CI) in total prescriptions | | | | | |
|---|---|---|---|---|---|---|---|---|
| Laryngitis | 1 | 1 | 100 (2.5–100) | 100 (2.5–100) | | | | 0 |
| | | | | 100 (2.5–100) | 0 | 0 | 0 | 0 | 0 |
| Bronchitis | 361 | 170 | 47.1 (41.8–52.4) | 91.2 (85.9–95) | | | | 8.8 (5–14.1) |
| | | | | 9.4 (5.5–14.8) | 39.4 (32–47.2) | 41.2 (33.7–49) | 1.2 (0.1–4.2) | 5.9 (2.9–10.6) | - |

CI, confidence interval.

The diseases were defined by International Classification of Diseases, 10th Revision, codes for the following conditions: Sinusitis [J01], pharyngitis [J02], tonsillitis [J03], laryngitis [J04, 05], and bronchitis [J20, 21, 22, 40, 209].

† Third-generation cephalosporins.

†† Total number of broad-spectrum antimicrobials and percentages of all prescribed antimicrobials.

## Discussion

We examined the antimicrobial prescriptions for patients diagnosed with ARTIs in the outpatient setting at five medical institutes in Japan in 2018. In comparison with those in previous studies in the literature based on health insurance claims data [9,11,13], the proportion of antimicrobial prescriptions in our cohort was much lower (less than 20%) in all age groups. While, broad-spectrum agents were the most frequently prescribed antimicrobials, as described in the previous literature. In the midst of AMS promotion to combat against the AMR, our findings could serve as an indicator for monitoring antimicrobial prescription for patients with ARTIs, which, we expect, can be useful data for health policymakers.

Contrary to our assumption, a low frequency of antimicrobial prescriptions for ARTIs was found in this study. Previous literature has highlighted the overuse of antimicrobials for the self-limiting diseases in Japan: approximately 60% in 2009 [9], 53.7% during 2013–2015 [11], and 31.7% during 2012–2017 [13]. The prescription rates seem to show a decreasing trend over time. The downward trend was well analyzed by Kimura *et al.*, who reported that the monthly antimicrobial prescription rate reduced by 19.2% from 2012 (34.4%) to 2017 (27.8%) [13]. In our study, the proportion of antimicrobial prescriptions was less than 20% in all age categories. According to a proposal from the European Surveillance of Antimicrobial Consumption Project Group, antimicrobial prescription rates for ARTIs should be less than 20% [19]. In contrast to published studies, our study directly investigated the medical records of the patients, and thus, the clinical diagnosis of ARTIs in our cohort would be much more reliable and the data is more likely to correctly reflect the present situation. Therefore, we assert that the antimicrobial prescriptions for ARTIs in our settings would be a status as per the guideline recommendations.

While the quantity of antimicrobial prescriptions has been optimized, there is room for improving the quality of prescriptions. ARTIs are, in principle, self-limiting diseases that require no antimicrobial treatment; however, broad-spectrum antimicrobials accounted for nearly 90% of the prescriptions. High antimicrobial prescription rates were similarly observed in every age group. Various observations have been made regarding the persistence of this unfavorable situation. An observational study in the United Kingdom showed that the higher the frequency of hospital visits, the higher were the antibiotic prescription rates and the more extensive was the usage of broad-spectrum antibiotics [20]. Authors in previous studies have discussed potential explanations, such as (i) antimicrobial prescribing is a time-sparing approach because physicians do not need to explain the diagnosis in detail and why they do

not require antimicrobials [21], (ii) patient's misperceptions derived from a patient's personal experience of being well treated with antimicrobials with good results [22], and (iii) expectation for covering diagnostic indeterminacy by prescribing broad-spectrum drugs. A potential influence of these factors may differ among medical situations, routine practices of individual clinicians, experiences and ages of patients, and societies with different cultural and healthcare backgrounds. A multidisciplinary approach is necessary across the area of expertise to make further progress in AMS for ARTIs.

Paradoxically, accessibility to hospitals can contribute to the increased number of antimicrobial prescriptions. In Japan, we have a universal health coverage system [23], which makes it possible to visit medical institutes frequently with ease, without intermediation. In fact, according to a previous study, the outpatient ARTIs visit rate was 990 per 1000 person-years; that is, every Japanese individual visits a hospital once a year due to ARTIs [11]. This rate was considered very high compared with those reported for other countries such as the United Kingdom (131 per 1000 patients per year) [24], Belgium (275 per 1000 patients), the Netherlands (141 per 1000 patients), and Sweden (132 per 1000 patients) [25]. Development of the universal health coverage system in Japan is praiseworthy when considering the achievement of equal access to medical care and improvement in public health. However, this progress, in turn, might have resulted in the over-prescription of antimicrobials in the past. Patients with ARTIs are usually satisfied with antimicrobial prescriptions by doctors [26]. Thus, to overcome this situation, social education, including that of parents of young children [11], should be continuously encouraged to improve the understanding of the population. In addition, further education and enlightenment of medical practitioners regarding this issue, irrespective of their specialties, is essential. Good accessibility to healthcare, on deeper introspection, might allow the adoption of a "delayed antibiotic prescription" strategy when considering the treatment of ARTIs [27]. Most cases of ARTIs are self-relieving without any antimicrobial treatment, although some patients may develop secondary complications. The number needed to treat for the prevention of secondary complications after common respiratory tract infections by antimicrobial prescription is reportedly over 4,000 [28]. Thus, we believe that the delayed antibiotics strategy can safely reduce the antibiotic use in patients with ARTIs, which should be more advocated among clinicians to promote AMS.

Antimicrobial prescriptions for respiratory infections other than the common cold should be discussed as well. For bronchitis, antimicrobial prescription rates were nearly 50% in each age group. Considering that bronchitis is mostly viral in nature, antimicrobials should be administered in 10% of such cases on average [5,6]; thus, a large amount of prescribed antibiotics might be potentially unnecessary. Recent guidelines also support this clinical stance of not prescribing antimicrobials for acute bronchitis, especially when patients present with various symptoms suggesting common cold [29,30]. Therefore, the high antimicrobial prescription for bronchitis should be rectified hereafter. Due to the small number of cases in our study, we were unable to discuss this issue regarding other types of ARTIs.

The present study had several strengths and limitations. Previous administrative data such as health insurance claims data are unreliable in nature, and it is unclear whether they truly reflect the actual state of antimicrobial prescription. In this study, more reliable data for individual diagnoses were extracted by directly accessing medical records. The limitations of this study should also be noted. First, despite the study being conducted in a multicenter setting, data were collected from five institutes in a limited area of the country; thus, it may not be accurately representative of the entire Japanese population. Second, although we referred to the medical records to obtain information on patients' symptoms, other variables such as the causative pathogen and medication allergies that could influence the decision for antimicrobial prescription have not been collected. Third, we could not determine whether the prescribed

drugs were for immediate use or delayed use. If prescribed for delayed use, such drugs may not drive AMR. Fourth, the ICD-10 codes given in the medical records might be labelled for convenience and in a manner that was not based on an actual diagnosis, so as to best fit their antimicrobial prescriptions. This could have been true in some cases, but cannot be reviewed at this point. Despite these limitations, our data was suggestive of a favorable practice of antimicrobial prescription for the common cold in Japan, heading for the achievement of the AMR action plan.

In conclusion, our findings from this clinical data-based study suggest a favorable reduction in the amount of antimicrobials prescribed for outpatients presenting with a common cold. However, the antimicrobial prescription for the common cold should be further reduced because it is caused by viral etiologies and resolves without specific treatment, usually in a few days. In addition, broad-spectrum antimicrobials are still prescribed for the common cold at high rates, spurring the need for future studies focusing on the choice of drugs. These findings may be useful for health policy makers as a benchmark for monitoring the effectiveness of AMS promotion strategies in Japan.

## Supporting information

**S1 Dataset.**
(XLSX)

## Acknowledgments

We would like to thank Editage (www.editage.jp) for assistance with editing this manuscript.

## Author Contributions

**Data curation:** Hideharu Hagiya.

**Formal analysis:** Tomoharu Ishida.

**Investigation:** Hiroyuki Honda, Yasuhiro Nakano, Hiroko Ogawa, Mikako Obika, Keigo Ueda, Hitomi Kataoka, Yoshihisa Hanayama.

**Project administration:** Fumio Otsuka.

**Supervision:** Hideharu Hagiya, Fumio Otsuka.

**Writing – original draft:** Tomoharu Ishida.

**Writing – review & editing:** Hideharu Hagiya, Yoshihisa Hanayama, Fumio Otsuka.

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
