## [Decision Letter · Decision Letter 0]

17 Aug 2021

PONE-D-21-23291

Antimicrobial Prescription Practice for Outpatients with Acute Respiratory Tract Infections: A Retrospective, Multicenter, Medical Record-based Study1

PLOS ONE

Dear Dr. Hagiya,

Thank you for submitting your manuscript to PLOS ONE. After careful consideration, we feel that it has merit but does not fully meet PLOS ONE’s publication criteria as it currently stands. Therefore, we invite you to submit a revised version of the manuscript that addresses the points raised during the review process.

Many thanks for submitting your manuscript to PLOS One

It was reviewed by two experts in the field, and they have recommended some modifications be made prior to acceptance

I therefore invite you to make these changes and to write a response to reviewers which will expedite revision upon resubmission

I wish you the best of luck with your modifications

Hope you are keeping safe and well in these difficult times

Thanks

Simon

We look forward to receiving your revised manuscript.

Kind regards,

Simon Clegg, PhD

Academic Editor

PLOS ONE

"No specific funding for this work was provided."

"None to declare."

Reviewers' comments:

Reviewer's Responses to Questions

**Comments to the Author**

1. Is the manuscript technically sound, and do the data support the conclusions?

Reviewer #1: Partly

Reviewer #2: Yes

2. Has the statistical analysis been performed appropriately and rigorously? 

Reviewer #1: No

Reviewer #2: Yes

3. Have the authors made all data underlying the findings in their manuscript fully available?

Reviewer #1: Yes

Reviewer #2: Yes

4. Is the manuscript presented in an intelligible fashion and written in standard English?

Reviewer #1: No

Reviewer #2: Yes

5. Review Comments to the Author

Reviewer #1: This study presents antibiotic prescription practices for acute respiratory tract infections across 7 medical institutions in Japan. Whilst this study is certainly timely and highly important, if affords further elaboration and statistical analyses, and more in-depth argumentation to better understand prescription practices across the 7 institutions and for various factors of interest.

Title:

• Comprehensive title, however, would remove the superscript (number 1) as it is just confusing.

• Would also consider slight modification. This study is evaluation antibiotic prescription practices specifically therefore I would be more specific in the title and avoid the term antimicrobial which encompasses all antimicrobial drugs, not solely antibiotics. Would therefore suggest slight rephrasing to: “Antibiotic Prescription Practices for Outpatients with Acute Respiratory Tract Infections: A Retrospective, Multicenter, Medical Record-based Study”

Background:

Overall background is structured well but can be made more concise and language should be revised. Please see my specific comments below:

• “In the guidelines, the use of antimicrobials for the treatment of ARTIs is not recommended [7,8].” This claim is untrue. Prescription of antibiotics for ARTIs is often unnecessary and not in accordance to guidelines, however there are circumstances, even in the guidelines where antibiotics are recommended, if not for immediate use, at least for delayed use.

• “However, in reality, antimicrobials are often prescribed in the clinical setting in Japan partially because of requests from patients or due to physicians’ anxiety.” What evidence do you have to support this claim? Is this anecdotal evidence or are there reports/studies that have found patient demand and physician anxiety to impact antibiotic prescription? Is it really a result of physician anxiety? Several other factors have been identified in the literature, including uncertainty, lack of access to diagnostic testing, perceived pressure from patients, knowledge and awareness, etc.

• “Approximately, in half of these visits antimicrobials were prescribed (532.4 per 1000 person-years; 99% CI, 531.6–533.3)”: provide the exact number.

• “Most of the antimicrobials prescribed for ARTIs were broad-spectrum oral formulations including cephalosporins (41.9%; third-generation cephalosporins, 97.3%), macrolides (32.8%), and fluoroquinolones (14.7%) [12].” The way data for cephalosporins are presented is very confusing. Please reframe.

• “Another Japanese study revealed that despite the decreasing trend, the issue of inappropriate antibiotic prescriptions for non-bacterial ARTIs persisted during the study period from 2012 to 2017 [13].” What were the prescription rates in this study and how do they compare to others? Did they asses guideline-concordant antibiotic prescription?

• “The proportion of antimicrobial prescriptions for ARTIs in Japan is reportedly higher than that in other countries [15–17].” Which countries? What are the rates there and how do they compare to Japanese data?

• Please better explain why insurance claims data is less credible than your methodology. Rather than just stating that it is less credible, explain why. Is it because it is biased towards more several ill patients? Do all patients file health insurance claims in Japan?

Methods:

• Are there any differences in the patient populations across the 7 different institutions? For example do some cater for urban whereas others cater for more rural populations? I would appreciate more information about the clinics. Are the capacities different? A busier clinic might have higher inappropriate prescribing rates just because doctors do not have the time to educate patients and think through their ordering practices as smaller clinics might.

• I would also like to know more about the prescription laws in Japan and who is authorized to prescribe. Are the prescriptions you are evaluating only provided by medical doctors? Or can nurse practitioners also prescribe and if so why not evaluate whether there are any statistically significant differences in the antibiotic prescribing patterns of the two, i.e. nurse practitioners versus medical doctors. You can also be more specific regarding what kind of medical doctors’ prescriptions you are assessing. Are they general practitioners? Are they specialists? Are they respiratory specialists?

• I think you need to explicitly motivate why you excluded all patients under 18 years of age.

• Why did you restrict your analysis to this small list of antibacterials? Are J01C antibacterials never prescribed in Japan for example? If so they should have definitely been captured as a broad-spectrum prescription. Why have they been excluded entirely? What about J01D antibacterials? You restricted your data capture to 3rd-gen cephalosporins and faropenem specifically, but what about other ‘other beta-lactam penicillins’ such as 1st and 2nd-gen cephalosporins?

• You collected data on age, sex, presence of upper or lower (cough and sputum expectoration) respiratory symptoms, clinical diagnosis defined by the ICD-10 codes, and antibiotic prescription. You also further categorized age groups. Why collect and categorize these data but then not use them for more in-depth analysis?

• You excluded patients with IV antibiotic therapy. Does that mean that you only included patients who received oral antibiotics? In that case, please specify that. Right now it seems like you included all patients with any antibiotic prescription for specific ICD-10 codes, except for IV antibiotic prescriptions.

• Statistical analysis is poorly written. Lacks detail. Even if analysis was descriptive, I would like to know which specific methods you used.

• Regarding your outcomes, this is the first time that we learn that you are also including other drugs prescribed for ARTIs, i.e. not just antibiotics. You should also motivate why your primary outcome is the common cold specifically.

• Were the prescriptions you analyzed all for immediate you? Can you differentiate whether they were for delayed use or not?

Results:

• Please specific the number of cases for all clinics (split the Niimi National Health Insurance Clinics into the 3 different clinics). This comment pertains figure 1 (which I think can be deleted), table 1 and the text. If you would like to keep that group of clinics together, then at least you need to explain why because in the methods you say that you analyze the prescription practices of 7 clinics and then you seemingly restrict to 5 which makes it confusing.

• First paragraph of the results can actually move to the methods section in my opinion, when you describe inclusion and exclusion criteria. Please also refrain from inserting number in the text. You should place them in brackets with “n=” before each number, e.g. “data of 3,940 patients were analyzed: Marugame Medical Center (n=996); Kasaoka City Hospital (n=1433)”, etc.

• Table 1: Please change title of the table (no informative). Also, lots of details are lacking. Make sure to provide them. For example, insert all percentages. Do not just present the ICD codes without explaining what they are; not all readers are familiar with them but will certainly be familiar with the description.

• No need to label categorized age with “adult, early elderly, late elderly”. Seems purely subjective labelling and unnecessary. Also please make sure not to refer to these labels in the text but rather you the age ranges; again for the sake of clarity. Please will not remember the age ranges for each of the labels you provided.

• You present your results using mean and SD. Are data normally distributed? Is this the correct summary statistic or should you have presented medians and IQRs? This is unclear since your description of your statistical analysis in the methods section is vague.

• Why collect data on symptoms but not present it here?

• In think you afford running statistical models on the data and not just keep it at a descriptive level. Are there statically significant prescription practices across the clinics, across age groups, across the various ICD-10 diagnosis? Are there statistically significant differences in prescribing practices if more than 1 clinical manifestation was involved?

• I find the way you have divided the total number of clinical manifestations to be confusing. You write all involved, 2 involved and >= 2 involved. Where there never just 1 clinical manifestation noted? And what’s the difference between all involved and >= 2 involved?

• In table 2, what does “others” refer to? Other antibacterials or other drugs such as symptomatic relief medications?

• In tables 2 and 3 why split the data by age if you do not statistically measure differences between the various age groups? Also, you now include 16 and 17 year old patients. Were they or were they not included? You also need to describe what kind of data you are presenting in these tables. In the legend you say that CI refers to confidence interval but it is not immediately obvious in the table which results you are referring to. Although it can be assumed, you need to specify.

Discussion:

• Whilst some good points are raised the discussion lacks depth and sometimes lacks flow. Please try re-writing it to bring arguments together in a more comprehensive manner.

• First paragraph refers to studies and previous literature but lacks references. And how do your data compare? How much lower that what other studies have found?

• Please consider sentence structure in the 1st paragraph. I would also be hesitant to claim that “broad spectrum agents were the most frequently prescribed antimicrobials” because from your methods your data capture seemed to be restricted to specific antibiotics/antibiotic classes and so any other narrow-spectrum antibiotics may have not been captured, as well as any other broad-spectrum antibacterials that for some reason were excluded from this study (not immediately clear as to why this decision was made).

• In the Kimura study, the trend decreased by 19.2%, but from what percentage? And what was the final percentage in 2017?

• There are a few studies from southern Europe that describe high broad-spectrum antibiotic use and uncertainty avoidance that I think can be referred to in the discussion.

• Delayed antibiotic prescription strategies are mentioned in one sentence. I would like you to expand upon this and explain why this strategy could be beneficial in your setting. Is this something that is commonly practiced or not? Is it worth investigating further?

• Something else you could consider noting is that doctors may have adjusted their diagnosis to best fit their ordering behavior. The advantage of your study design is that data were pulled retrospectively and that doctors were not aware that their prescription practices were to be analyzed.

• Finally, I think the fact that a good percentage of the patients received antibiotics for the common cold should not be overlooked. The conclusion to me is more positive than it should be. Whilst the prescription rates may be lower than other settings, they are not low enough, and certainly not for the common cold where prescription should be down at 0%. It is good that you highlight however that broad-spectrum antibiotic use is high and needs to be addressed.

General comments:

• Language: Please take the time to do an extensive review of the manuscript’s language. Whilst the manuscript is well-written overall, it can be made more concise and there are some grammatical and sentence-structure issues that need correcting before publication. Arguments in the discussion can also flow better. I would also avoid sweeping statements and using words such as “menace”.

Reviewer #2: • The data from this study suggest that a lower percentage of ARTI cases (common cold) are given antibiotics (12%) than in 2018, but that ~90% of these are broad spectrum. This corresponds with your conclusions, which report this downwards trend and also the discussion of how to reduce broad spectrum usage

• The only analysis you use is confidence intervals, more could have been made of the data as there was no use of tests for identifying differences between groups, e.g. age groups, types of drugs used and for what infection type, which could have identified some more, perhaps interesting, results. If you had data from previous surveys, this could have also been compared

• According to your declaration, you have made all data available

• You have written this in clear and understandable language, although whilst the discussion is clear that ARTIs are mainly viral, this could be made clearer in the introduction

6. PLOS authors have the option to publish the peer review history of their article (what does this mean?). If published, this will include your full peer review and any attached files.

Reviewer #1: No

---

## [Author Response · Author response to Decision Letter 0]

30 Sep 2021

29th/September/2021

Dear Prof. Simon Clegg, PhD

Academic Editor

PLOS ONE

Ref: PONE-D-21-23291-R1

Antimicrobial Prescription Practice for Outpatients with Acute Respiratory Tract Infections: A Retrospective, Multicenter, Medical Record-based Study

We hereby resubmit our above-named manuscript for reconsideration for publication in PLOS ONE. We have carefully considered all of the enclosed comments and addressed them as thoroughly as possible. Point-by-point responses to the reviewers’ comments are given below. The corrected sentences are noted with track changes in the revised version. 

We hope you will now find our revised manuscript finally acceptable for publication in PLOS ONE.

Sincerely yours,

Hideharu Hagiya, M.D., Ph.D.

Department of General Medicine, Okayama University Graduate School of Medicine, Dentistry and Pharmaceutical Sciences, 2-5-1 Shikata-cho, Kita-ku, Okayama 700-8558, Japan 

Tel: +81-86-235-7342 Fax: +81-86-235-7345

E-mail: hagiya@okayama-u.ac.jp

 

Comment from Reviewer #1

This study presents antibiotic prescription practices for acute respiratory tract infections across 7 medical institutions in Japan. Whilst this study is certainly timely and highly important, if affords further elaboration and statistical analyses, and more in-depth argumentation to better understand prescription practices across the 7 institutions and for various factors of interest.

Response

We greatly appreciate your effort to review our study. We have provided point-by-point comments below.

Title:

• Comprehensive title, however, would remove the superscript (number 1) as it is just confusing.

Response

As per your indication, we have deleted the superscript.

• Would also consider slight modification. This study is evaluation antibiotic prescription practices specifically therefore I would be more specific in the title and avoid the term antimicrobial which encompasses all antimicrobial drugs, not solely antibiotics. Would therefore suggest slight rephrasing to: “Antibiotic Prescription Practices for Outpatients with Acute Respiratory Tract Infections: A Retrospective, Multicenter, Medical Record-based Study”

Response

Thank you for your advice. In general, it is usual to use the term “antimicrobials”, rather than “antibiotics”, in this specialized area of infectious disease. Also, it is obvious that we are dealing with “antibiotics” in this study when readers go through the manuscript. Thus, we would like to remain “antimicrobials” as it is.

Background:

Overall background is structured well but can be made more concise and language should be revised. Please see my specific comments below:

• “In the guidelines, the use of antimicrobials for the treatment of ARTIs is not recommended [7,8].” This claim is untrue. Prescription of antibiotics for ARTIs is often unnecessary and not in accordance to guidelines, however there are circumstances, even in the guidelines where antibiotics are recommended, if not for immediate use, at least for delayed use.

Response

Thank you for your advice. According to your comment, we have changed the sentence as follows; According to the guidelines, prescription of antimicrobials is often unnecessary for ARTIs treatment, although there are actually cases that require a delayed prescription [7,8] (Line 67-68)

• “However, in reality, antimicrobials are often prescribed in the clinical setting in Japan partially because of requests from patients or due to physicians’ anxiety.” What evidence do you have to support this claim? Is this anecdotal evidence or are there reports/studies that have found patient demand and physician anxiety to impact antibiotic prescription? Is it really a result of physician anxiety? Several other factors have been identified in the literature, including uncertainty, lack of access to diagnostic testing, perceived pressure from patients, knowledge and awareness, etc.

Response

This sentence is just an anecdotal comment from us, without a scientific analysis. Thus, we have changed this sentence as follows; However, in reality, we, as Japanese clinicians, frequently witness antimicrobials being prescribed in outpatient settings, partially because of requests from patients or due to physicians’ anxiety. (Line 68-70)

• “Approximately, in half of these visits antimicrobials were prescribed (532.4 per 1000 person-years; 99% CI, 531.6–533.3)”: provide the exact number.

Response

These numbers are the exactly given in the literature. Please go through it. We have modified the sentence so as it is more understandable; antimicrobials were prescribed in half of these visits (532.4 per 1000 person-years; 99% CI, 531.6–533.3) [11] (Line 77-78)

• “Most of the antimicrobials prescribed for ARTIs were broad-spectrum oral formulations including cephalosporins (41.9%; third-generation cephalosporins, 97.3%), macrolides (32.8%), and fluoroquinolones (14.7%) [12].” The way data for cephalosporins are presented is very confusing. Please reframe.

Response

We rephrased the corresponding sentence as follows; Most of the antimicrobials prescribed for ARTIs were broad-spectrum oral formulations including cephalosporins (41.9%, of which third-generation cephalosporins accounted for 97.3%), macrolides (32.8%), and fluoroquinolones (14.7%) [12] (Line 78-80).

• “Another Japanese study revealed that despite the decreasing trend, the issue of inappropriate antibiotic prescriptions for non-bacterial ARTIs persisted during the study period from 2012 to 2017 [13].” What were the prescription rates in this study and how do they compare to others? Did they assess guideline-concordant antibiotic prescription?

• “The proportion of antimicrobial prescriptions for ARTIs in Japan is reportedly higher than that in other countries [15–17].” Which countries? What are the rates there and how do they compare to Japanese data?

Response

The monthly antibiotics prescription rate was 31.65 per 100 consultations for nonbacterial ARTIs. They have just retrospectively collected data on the antibiotics prescribing and thus the study was not an evaluation for the Guideline compliance rate.

The prescribing rate cannot be simply compared to other preceding studies since the claims data is not identical each other. However, we introduced the results of previous study from the United States as examples.

We have changed the sentences as follows; Despite the higher antimicrobial prescription rates in Japan, downward trends have been reported. A retrospective, observational study using longitudinal, administrative claims data revealed that a mean monthly antimicrobial prescription rate for nonbacterial-ARTIs was 31.65 per 100 visits between April 2012 and June 2017 [14]. The antimicrobial prescription rate decreased by 19.2% during the study period; however, there was no remarkable trend change compared to other countries. For instance, previous national data in the United States (1995 to 2006) suggested that ARTIs-associated antimicrobial prescriptions decreased by 36% among children younger than 5 years and by 18% among persons aged 5 years or older [15]. According to another analysis of nationally representative data in the United States (2000 to 2010), antimicrobial prescription for ARTIs decreased by 57% among children and adolescents (<18 years) and 38% among adults (18 to 64 years), although there was no certain trend among those aged ≥65 years [16]. Thus, it is possible that the decrease in the proportion of antimicrobial prescriptions for ARTIs in Japan may be further accelerated. (Line 84-95)

• Please better explain why insurance claims data is less credible than your methodology. Rather than just stating that it is less credible, explain why. Is it because it is biased towards more several ill patients? Do all patients file health insurance claims in Japan?

Response

The health insurance claims data lacks credibility because it is accumulated without clinical records. We have added this sentence in Line 97-99.

All the clinical data on Japanese patients is not filed at all.

Methods:

• Are there any differences in the patient populations across the 7 different institutions? For example, do some cater for urban whereas others cater for more rural populations? I would appreciate more information about the clinics. Are the capacities different? A busier clinic might have higher inappropriate prescribing rates just because doctors do not have the time to educate patients and think through their ordering practices as smaller clinics might.

Response

All these institutes are located in the rural areas, and the patient populations are almost identical. The first four institutes (Marugame, Kasaoka, Tamano, and Kaneda) are regional general hospitals with inpatient beds, while the Niimi National Health Insurance Clinics are no-bedded outpatient clinics. (Line 108-111)

• I would also like to know more about the prescription laws in Japan and who is authorized to prescribe. Are the prescriptions you are evaluating only provided by medical doctors? Or can nurse practitioners also prescribe and if so why not evaluate whether there are any statistically significant differences in the antibiotic prescribing patterns of the two, i.e. nurse practitioners versus medical doctors. You can also be more specific regarding what kind of medical doctors’ prescriptions you are assessing. Are they general practitioners? Are they specialists? Are they respiratory specialists?

Response

In Japan, only medical doctors are authorized to prescribe antimicrobials, but not nurse practitioners. The antimicrobial prescriptions included in this study were not limited to either general practitioners or any organ specialists. (Line 128-129)

• I think you need to explicitly motivate why you excluded all patients under 18 years of age.

Response

In Japan, it is usual to deem individuals aged >18 years as adults. Thus we excluded those whose ages were 18 years or less.

• Why did you restrict your analysis to this small list of antibacterials? Are J01C antibacterials never prescribed in Japan for example? If so they should have definitely been captured as a broad-spectrum prescription. Why have they been excluded entirely? What about J01D antibacterials? You restricted your data capture to 3rd-gen cephalosporins and faropenem specifically, but what about other ‘other beta-lactam penicillins’ such as 1st and 2nd-gen cephalosporins?

Response

I think the reviewer misunderstand ICD-10 codes and ATC classification. In the 2nd paragraph of the Method section, we refer to the classification of ICD-10 codes to define ARTIs by referring to the previous, related studies [references 6,11,17]. For the classification of antibiotics, we referred to other reports [references 15, 18]. (Line 133-135)

• You collected data on age, sex, presence of upper or lower (cough and sputum expectoration) respiratory symptoms, clinical diagnosis defined by the ICD-10 codes, and antibiotic prescription. You also further categorized age groups. Why collect and categorize these data but then not use them for more in-depth analysis?

Response

We have utilized the collected data of age to stratified the age groups. Presence of upper or lower (cough and sputum expectoration) respiratory symptoms were used to confirm whether the diagnosis given by the ICD-10 codes was clinically endorsed or not. We did not stratify the data by sex; however, otherwise, we have used the collected data appropriately.

• You excluded patients with IV antibiotic therapy. Does that mean that you only included patients who received oral antibiotics? In that case, please specify that. Right now it seems like you included all patients with any antibiotic prescription for specific ICD-10 codes, except for IV antibiotic prescriptions.

Response

We excluded those who received intravenous antimicrobial therapy to include patients prescribed with oral antimicrobials alone. (Line 158-159)

> Right now it seems like you included all patients with any antibiotic prescription for specific ICD-10 codes, except for IV antibiotic prescriptions.

Yes, that is true for our study.

• Statistical analysis is poorly written. Lacks detail. Even if analysis was descriptive, I would like to know which specific methods you used.

Response

We have totally amended the statistical method parts. Please go through the revised paragraph (Line 137-158)

• Regarding your outcomes, this is the first time that we learn that you are also including other drugs prescribed for ARTIs, i.e. not just antibiotics. You should also motivate why your primary outcome is the common cold specifically.

Response

Common cold is the representative of ARTIs, which is known to be caused by viral etiologies in most of the cases. Actually our main concern is the antibiotics prescription rates for the common cold. We are sorry for the insufficient explanation. We have amended the final sentence of the Introduction part as follows; The present study aimed to determine the proportion of antimicrobial prescriptions for ARTIs, especially for the common cold, by directly examining the medical records. (Line 100-101)

• Were the prescriptions you analyzed all for immediate use? Can you differentiate whether they were for delayed use or not?

Response

This is important issue, however, unfortunately, it is impossible to determine the prescribed antibiotics were for immediate use or delayed use. We have added this sentence at Line 310-311.

Results:

• Please specific the number of cases for all clinics (split the Niimi National Health Insurance Clinics into the 3 different clinics). This comment pertains figure 1 (which I think can be deleted), table 1 and the text. If you would like to keep that group of clinics together, then at least you need to explain why because in the methods you say that you analyze the prescription practices of 7 clinics and then you seemingly restrict to 5 which makes it confusing.

Response

Thank you for your advice. As the total number of patients in the Niimi National Health Insurance Clinics is only 51. Thus there is no merit to spilt data more in detail. We therefore explained that we have collected data from “five” clinics, not seven clinics, through the manuscript.

• First paragraph of the results can actually move to the methods section in my opinion, when you describe inclusion and exclusion criteria. Please also refrain from inserting number in the text. You should place them in brackets with “n=” before each number, e.g. “data of 3,940 patients were analyzed: Marugame Medical Center (n=996); Kasaoka City Hospital (n=1433)”, etc.

Response

It is natural for us to place this first paragraph as it is. We expect your understanding. While, the way describing the number of cases were amended as recommended. 

• Table 1: Please change title of the table (no informative). Also, lots of details are lacking. Make sure to provide them. For example, insert all percentages. Do not just present the ICD codes without explaining what they are; not all readers are familiar with them but will certainly be familiar with the description.

Response

The title was changed to “Numbers and percentages of background data and diagnosis of the cases in each medical institute by age groups.”. Also, we have provided percentages in each figure. 

The ICD-10 codes are explained as follows in the Table foot note; The International Classification of Diseases (ICD-10) codes were endorsed in May 1990 by the Forty-third World Health Assembly to develop the diagnostic classification standard for all clinical and research purposes.

• No need to label categorized age with “adult, early elderly, late elderly”. Seems purely subjective labelling and unnecessary. Also please make sure not to refer to these labels in the text but rather you the age ranges; again for the sake of clarity. Please will not remember the age ranges for each of the labels you provided.

Response

We unlabeled the age groups. As per the advice, we deleted the labelling from the text entirely.

• You present your results using mean and SD. Are data normally distributed? Is this the correct summary statistic or should you have presented medians and IQRs? This is unclear since your description of your statistical analysis in the methods section is vague.

Response

We appreciate your comment. We cannot demonstrate all the data is normally distributed in this study. Thus, we summarized the data with using median and IQRs, instead of mean and SD. Additionally, we have newly made it clear that continuous variables were summarized with median and IQRs in the Method section. (Line 142-143)

• Why collect data on symptoms but not present it here?

Response

We applied the symptoms data in Table 2 to confirm that patients diagnosed with common cold had two or more or organ-related symptoms.

• In think you afford running statistical models on the data and not just keep it at a descriptive level. Are there statically significant prescription practices across the clinics, across age groups, across the various ICD-10 diagnosis? Are there statistically significant differences in prescribing practices if more than 1 clinical manifestation was involved?

Response

We appreciate your recommendation. In this study, we are not interested in the comparison of antimicrobial prescription among the five medical institutes and among the cases with different ICD-10 diagnosis. Our primary aim of the present study is to investigate the proportion of antimicrobial prescriptions for ARTIs, especially the common cold. We thus additionally performed a statistical analysis for such purpose. Strength of this study is to collect clinical data endorsing the diagnosis of common cold, therefore, we stratified the eligible cases by the numbers of clinical manifestations suggesting common cold; “Three respiratory regions”, “Two respiratory regions”, and “Two or more respiratory regions”, as have done in the first-submitted manuscript. The results are given in the newly-formatted Table 2 and the second paragraph of the Discussion.

• I find the way you have divided the total number of clinical manifestations to be confusing. You write all involved, 2 involved and >= 2 involved. Where there never just 1 clinical manifestation noted? And what’s the difference between all involved and >= 2 involved?

Response

Sorry for the confusing description. As written above, we have renamed the term as either of “Three respiratory regions”, “Two respiratory regions”, and “Two or more respiratory regions”. Also, the explanations are amended as in the Table footnote as follows;

*"Three respiratory regions” denotes that clinical manifestations of all three distinct respiratory tract regions (nasal, pharyngo-laryngeal, and bronchial) were described in the medical records. **"Two respiratory regions” and ***"Two or more respiratory regions” denote that as it appears, clinical manifestations of each of 

the two and two or more of the respiratory tract regions were described.

Because we would like to confirm there were clinical manifestations relating to ARTIs in the eligible patients in this study, we were interested in the numbers of organs (either each of nasal, pharyngo-laryngeal, or bronchial areas). Patients with “Three respiratory regions” were those who manifested all the three distinct respiratory tract regions. While, patients with " Two or more respiratory regions” we those who manifested two or three distinct respiratory tract regions.

• In table 2, what does “others” refer to? Other antibacterials or other drugs such as symptomatic relief medications?

Response

“others” referred to the other antimicrobials. We have amended it as appropriate.

• In tables 2 and 3 why split the data by age if you do not statistically measure differences between the various age groups? Also, you now include 16 and 17 year old patients. Were they or were they not included? You also need to describe what kind of data you are presenting in these tables. In the legend you say that CI refers to confidence interval but it is not immediately obvious in the table which results you are referring to. Although it can be assumed, you need to specify.

Response

As written above, we performed the statistical analysis to compare the antimicrobial prescription among the age group (see the new Table 2). 

We included those aged 18 years and more in this study, not 16 and 17 year-old patients. The age category was appropriately revised as “18-64 years”. 

We have improved the Table contents considering the readability.

Discussion:

• Whilst some good points are raised the discussion lacks depth and sometimes lacks flow. Please try re-writing it to bring arguments together in a more comprehensive manner.

Response

Thank you for your comment on our Discussion. We have amended it with caution so as it conveys meaningful points to readers, with better flow.

• First paragraph refers to studies and previous literature but lacks references. And how do your data compare? How much lower that what other studies have found?

Response

Our data showed antimicrobial prescribing rates for common cold were less than 20% in all the age group, which was much lower than previous studies based on health insurance claims data (Reference 9, 11, 13). We have provided the reference numbers as appropriate. Please see the second paragraph of the Discussion where the details are given.

• Please consider sentence structure in the 1st paragraph. I would also be hesitant to claim that “broad spectrum agents were the most frequently prescribed antimicrobials” because from your methods your data capture seemed to be restricted to specific antibiotics/antibiotic classes and so any other narrow-spectrum antibiotics may have not been captured, as well as any other broad-spectrum antibacterials that for some reason were excluded from this study (not immediately clear as to why this decision was made).

Response

As already described in the Data collection paragraph of the Method section, we considered third-generation cephalosporins, macrolides, fluoroquinolones, and faropenem as broad-spectrum antimicrobials, while deemed penicillins as narrow-spectrum antimicrobials, according to previous studies [reference 15,18]. (Line 133-135) As clearly shown in the Table 2, if prescribed, the antibiotics were those included in the broad-spectrum drugs. We did not restrict data into these antibiotics alone, and opened all the data collected from the survey. Truly, the broad-spectrum antibiotics, such as third-generation cephalosporins, macrolides, fluoroquinolones, and faropenem were prescribed with high proportions. See the newly-given Table 2.

• In the Kimura study, the trend decreased by 19.2%, but from what percentage? And what was the final percentage in 2017?

Response

According to the study, the antibiotics prescriptions for nonbacterial ARTIs in 2012 and 2017 were 34.4 and 27.8, respectively. (Line 252-254)

• There are a few studies from southern Europe that describe high broad-spectrum antibiotic use and uncertainty avoidance that I think can be referred to in the discussion.

Response

We have already cited many related articles and the total number of the citation is up to 30. We thus consider it is enough to cite previous studies at the present manuscript.

• Delayed antibiotic prescription strategies are mentioned in one sentence. I would like you to expand upon this and explain why this strategy could be beneficial in your setting. Is this something that is commonly practiced or not? Is it worth investigating further?

Response

Thank you for your comment and advice. We have expanded the discussion as in Line 287-2298 by incorporating the data on the number needed to treat for ARTIs by antibiotics.

• Something else you could consider noting is that doctors may have adjusted their diagnosis to best fit their ordering behavior. The advantage of your study design is that data were pulled retrospectively and that doctors were not aware that their prescription practices were to be analyzed.

Response

Thank you for reminding us of this point. We have added this as the 4th limitation of this study; Fourth, the ICD-10 codes given in the medical records might be labelled for convenience and in a manner that was not based on an actual diagnosis, so as to best fit their antimicrobial prescriptions. This could have been true in some cases, but cannot be reviewed at this point. (Line 311-314)

• Finally, I think the fact that a good percentage of the patients received antibiotics for the common cold should not be overlooked. The conclusion to me is more positive than it should be. Whilst the prescription rates may be lower than other settings, they are not low enough, and certainly not for the common cold where prescription should be down at 0%. It is good that you highlight however that broad-spectrum antibiotic use is high and needs to be addressed.

Response

Thank you for understanding our conclusion. As mentioned, the antimicrobial prescription for common cold should be much more reduced. We feel that now is on the way to a positive future with such good practice. We have amended the conclusion paragraph according to your advice as follows;

In conclusion, our findings from this clinical data-based study suggest a favorable reduction in the amount of antimicrobials prescribed for outpatients presenting with a common cold. However, the antimicrobial prescription for the common cold should be further reduced because it is caused by viral etiologies and resolves without specific treatment, usually in a few days. In addition, broad-spectrum antimicrobials are still prescribed for the common cold at high rates, spurring the need for future studies focusing on the choice of drugs. These findings may be useful for health policy makers as a benchmark for monitoring the effectiveness of AMS promotion strategies in Japan.

General comments:

• Language: Please take the time to do an extensive review of the manuscript’s language. Whilst the manuscript is well-written overall, it can be made more concise and there are some grammatical and sentence-structure issues that need correcting before publication. Arguments in the discussion can also flow better. I would also avoid sweeping statements and using words such as “menace”.

Response

Before resubmission, we have again made the manuscript being checked by an English-native. Please again go through the text whether it has reached an acceptable level in terms of English grammar. 

 

Comment from Reviewer #2

Comment #1

The data from this study suggest that a lower percentage of ARTI cases (common cold) are given antibiotics (12%) than in 2018, but that ~90% of these are broad spectrum. This corresponds with your conclusions, which report this downwards trend and also the discussion of how to reduce broad spectrum usage

Response #1

Thank you for your comment. We appreciate your understanding for our study conclusion. Beyond our expectation, the antimicrobial prescribing rates for outpatients with ARTIs were limited to 12.7%. However, as in the conclusion, the broad-spectrum antimicrobials are still frequently prescribed, which urges us to promote further antimicrobial stewardship in the outpatient setting.

Comment #2

The only analysis you use is confidence intervals, more could have been made of the data as there was no use of tests for identifying differences between groups, e.g. age groups, types of drugs used and for what infection type, which could have identified some more, perhaps interesting, results. If you had data from previous surveys, this could have also been compared.

Response #2

Thank you for your advice. As per the recommendation by you and another reviewer, we additionally performed the statistical analysis to compare the antimicrobial prescription among the age groups. The results are given in the new Table 2. This time, we are not intended to antimicrobial prescribing among each medical institute. 

Comment #3

According to your declaration, you have made all data available

Response #3

Our data is available if requesting to a corresponding author.

Comment #4

You have written this in clear and understandable language, although whilst the discussion is clear that ARTIs are mainly viral, this could be made clearer in the introduction

Response #4

We have already provided the fact that the majority of the ARTIs is virally caused (Line 65-66). 

Thank you for your review.

---

## [Decision Letter · Decision Letter 1]

19 Oct 2021

PONE-D-21-23291R1Antimicrobial Prescription Practices for Outpatients with Acute Respiratory Tract Infections: A Retrospective, Multicenter, Medical Record-based StudyPLOS ONE

Dear Dr. Hagiya,

Thank you for submitting your manuscript to PLOS ONE. After careful consideration, we feel that it has merit but does not fully meet PLOS ONE’s publication criteria as it currently stands. Therefore, we invite you to submit a revised version of the manuscript that addresses the points raised during the review process.

Many thanks for submitting your manuscript to PLOS One

It was reviewed by two experts in the field, and they have recommended some very minor modifications be made prior to acceptance

I therefore invite you to make these changes and to write a response to reviewers which will expedite revision upon resubmission

I wish you the best of luck with your modifications

Hope you are keeping safe and well in these difficult times

Thanks

Simon

We look forward to receiving your revised manuscript.

Kind regards,

Simon Clegg, PhD

Academic Editor

PLOS ONE

Journal Requirements:

Reviewers' comments:

Reviewer's Responses to Questions

**Comments to the Author**

1. If the authors have adequately addressed your comments raised in a previous round of review and you feel that this manuscript is now acceptable for publication, you may indicate that here to bypass the “Comments to the Author” section, enter your conflict of interest statement in the “Confidential to Editor” section, and submit your "Accept" recommendation.

Reviewer #1: (No Response)

Reviewer #3: (No Response)

2. Is the manuscript technically sound, and do the data support the conclusions?

Reviewer #1: Yes

Reviewer #3: Yes

3. Has the statistical analysis been performed appropriately and rigorously? 

Reviewer #1: Yes

Reviewer #3: Yes

4. Have the authors made all data underlying the findings in their manuscript fully available?

Reviewer #1: Yes

Reviewer #3: Yes

5. Is the manuscript presented in an intelligible fashion and written in standard English?

Reviewer #1: Yes

Reviewer #3: Yes

6. Review Comments to the Author

Reviewer #1: Thank you for addressing my comments thoroughly and appropriately, and for giving me the opportunity to review this paper. I have just two remaining minor comments regarding the tables.

For table 1, as per my previous comment, please include the full names of the ICD-10 codes in the footnote, i.e. acute nasopharyngitis (J00), acute sinusitis (J01), acute pharyngitis (J02), acute tonsillitis (J03), acute laryngitis and tracheitis (J04), etc.... For someone not too familiar with the codes, it is important that that information is directly available in the table.

In table 3, the categorization of clinical manifestations is still ambiguous to me. Whats the difference between two or more respiratory regions and three respiratory regions? Were there no patients in any of the age groups that presented with only 1 respiratory region affected? What is the miscellaneous column representing? I don't see any data presented for it in this table (neither in table 4). Why not remove that column from both tables?

Reviewer #3: This is a very nicely written and interesting manuscript, for which I only have very minor comments

Line 40- 50.1% of cases (reword)

Line 68-69- In approximately half of these visits …. (reword)

Line 91- I think doing a separate study on children would be of interest here too

Line 139-140- this reads slightly awkwardly and may read better as- There were no notable differences in the proportions of antimicrobial prescriptions in cases involving infection in all, two or one region of the respiratory tract (16.7%, 18.2% and 17.7% respectively).

Line 175- the higher the frequency of … (reword)

Line 179- how much of this is an expectation due to age? So when a younger 70 year old went to the doctors, they would always get antibiotics, but the shift in thinking may affect this?

Line 222- perhaps a comparison of this data to administration data maybe useful?

Table 1- I think having the full data in for the J code maybe useful here

But overall, a very good manuscript and a pleasure to read

7. PLOS authors have the option to publish the peer review history of their article (what does this mean?). If published, this will include your full peer review and any attached files.

Reviewer #3: No

---

## [Author Response · Author response to Decision Letter 1]

20 Oct 2021

20th/October/2021

Dear Prof. Simon Clegg, PhD

Academic Editor

PLOS ONE

Ref: PONE-D-21-23291-R2

Antimicrobial Prescription Practice for Outpatients with Acute Respiratory Tract Infections: A Retrospective, Multicenter, Medical Record-based Study

We hereby resubmit our above-named manuscript for reconsideration for publication in PLOS ONE. We have carefully considered all of the enclosed comments and addressed them as thoroughly as possible. Point-by-point responses to the reviewers’ comments are given below. The corrected sentences are noted with track changes in the revised version. 

We hope you will now find our revised manuscript finally acceptable for publication in PLOS ONE.

Sincerely yours,

Hideharu Hagiya, M.D., Ph.D.

Department of General Medicine, Okayama University Graduate School of Medicine, Dentistry and Pharmaceutical Sciences, 2-5-1 Shikata-cho, Kita-ku, Okayama 700-8558, Japan 

Tel: +81-86-235-7342 Fax: +81-86-235-7345

E-mail: hagiya@okayama-u.ac.jp

 

Comment from Reviewer #1

Reviewer #1

Thank you for addressing my comments thoroughly and appropriately, and for giving me the opportunity to review this paper. I have just two remaining minor comments regarding the tables.

For table 1, as per my previous comment, please include the full names of the ICD-10 codes in the footnote, i.e. acute nasopharyngitis (J00), acute sinusitis (J01), acute pharyngitis (J02), acute tonsillitis (J03), acute laryngitis and tracheitis (J04), etc.... For someone not too familiar with the codes, it is important that that information is directly available in the table.

Response

We totally agree with your advice and provided the required information in the footnote of the Table 1.

In table 3, the categorization of clinical manifestations is still ambiguous to me. Whats the difference between two or more respiratory regions and three respiratory regions? Were there no patients in any of the age groups that presented with only 1 respiratory region affected? What is the miscellaneous column representing? I don't see any data presented for it in this table (neither in table 4). Why not remove that column from both tables?

Response

Thank you for your comment again. Usually, acute respiratory tract infections (ARTIs) causes a variety of manifestations, since pathogens infects with a broad part of the respiratory tracts (nasal, throat, and bronchi). Therefore, we are interested whether the respiratory tracts of the patients were widely involved or not. “Three respiratory regions” means all these respiratory tracts are involved. “Two respiratory regions” denotes two of the three respiratory tracts are involved. “Two or more respiratory regions” is a total of these cases. The last pattern may not be necessarily needed; however, we are intended to make it clear all these patient patterns. 

Patients with only 1 respiratory region were excluded from this Table, because they were possibly infected with bacterial pathogens that require antimicrobial treatment, to which we are not interested in this study. Similarly, data for miscellaneous cases is not main purpose of this study, and thus, we dared not present such cases in the table. Hoping your understanding.

 Thank you for your review.

Reviewer #3

This is a very nicely written and interesting manuscript, for which I only have very minor comments

Response

We greatly appreciate your comment. Amendments were given appropriately as per your advice.

Line 40- 50.1% of cases (reword)]

Response

This was revised appropriately.

Line 68-69- In approximately half of these visits …. (reword)

Response

This was revised appropriately.

Line 91- I think doing a separate study on children would be of interest here too

Response

Thank you for your comment. We agree that another study specifically targeting children is warranted in a future.

Line 139-140- this reads slightly awkwardly and may read better as- There were no notable differences in the proportions of antimicrobial prescriptions in cases involving infection in all, two or one region of the respiratory tract (16.7%, 18.2% and 17.7% respectively).

Response

The corresponding sentence is rewritten. Please go through the Line 194-199.

Line 175- the higher the frequency of … (reword)

Response

This was revised appropriately.

Line 179- how much of this is an expectation due to age? So when a younger 70 year old went to the doctors, they would always get antibiotics, but the shift in thinking may affect this?

Response

It is very interesting point to be addressed. Patient’s age potentially affects their behaviors for seeking antibiotics. However, it cannot be estimated in our data. Instead, we referred that various background factors may influence on the antimicrobial prescribing as follows; A potential influence of these factors may differ among medical situations, routine practices of individual clinicians, experiences and ages of patients, and societies with different cultural and healthcare backgrounds.

Line 222- perhaps a comparison of this data to administration data maybe useful?

Response

We appreciate your recommendation. It might provide useful information, but this time, it is beyond our purpose. In future study, we will challenge for that.

Table 1- I think having the full data in for the J code maybe useful here

Response

We believe the present data of ICD-10 codes given in the Table 1 is enough to identify the respiratory diseases. Other similar articles also described their ICD-10 codes as we do.

But overall, a very good manuscript and a pleasure to read.

Response

Thank you for your review in this difficult time.

---

## [Editor Report · Decision Letter 2]

25 Oct 2021

Antimicrobial Prescription Practices for Outpatients with Acute Respiratory Tract Infections: A Retrospective, Multicenter, Medical Record-based Study

PONE-D-21-23291R2

Dear Dr. Hagiya,

We’re pleased to inform you that your manuscript has been judged scientifically suitable for publication and will be formally accepted for publication once it meets all outstanding technical requirements.

Kind regards,

Simon Clegg, PhD

Academic Editor

PLOS ONE

Additional Editor Comments:

Many thanks for resubmitting your manuscript to PLOS One

As you have addressed all the comments and the manuscript reads well, I have recommended it for publication

You should hear from the Editorial Office shortly.

It was a pleasure working with you and I wish you the best of luck for your future research

Hope you are keeping safe and well in these difficult times

Thanks

Simon

---

## [Editor Report · Acceptance letter]

3 Nov 2021

PONE-D-21-23291R2 

Antimicrobial Prescription Practices for Outpatients with Acute Respiratory Tract Infections: A Retrospective, Multicenter, Medical Record-based Study 

Dear Dr. Hagiya:

I'm pleased to inform you that your manuscript has been deemed suitable for publication in PLOS ONE. Congratulations! Your manuscript is now with our production department. 

Kind regards, 

on behalf of

Dr. Simon Clegg 

Academic Editor

PLOS ONE